# Venestatin from parasitic helminths interferes with receptor for advanced glycation end products (RAGE)-mediated immune responses to promote larval migration

Daigo Tsubokawa[1,2]*, Taisei Kikuchi[3], Jae Man Lee[4], Takahiro Kusakabe[5], Yasuhiko Yamamoto[6], Haruhiko Maruyama[3]

1 Department of Molecular and Cellular Parasitology, Kitasato University Graduate School of Medical Sciences, Sagamihara, Japan, 2 Department of Parasitology and Tropical Medicine, Kitasato University School of Medicine, Sagamihara, Japan, 3 Division of Parasitology, Department of Infectious Diseases, Faculty of Medicine, University of Miyazaki, Miyazaki, Japan, 4 Laboratory of Creative Science for Insect Industries, Kyushu University Graduate School of Bioresource and Bioenvironmental Sciences, Fukuoka, Japan, 5 Laboratory of Insect Genome Science, Kyushu University Graduate School of Bioresource and Bioenvironmental Sciences, Fukuoka, Japan, 6 Department of Biochemistry and Molecular Vascular Biology, Kanazawa University Graduate School of Medical Sciences, Kanazawa, Japan

* dtsubo@med.kitasato-u.ac.jp

**Data Availability Statement:** All relevant data are within the manuscript and its Supporting Information files.

## Abstract

Parasitic helminths can reside in humans owing to their ability to disrupt host protective immunity. Receptor for advanced glycation end products (RAGE), which is highly expressed in host skin, mediates inflammatory responses by regulating the expression of pro-inflammatory cytokines and endothelial adhesion molecules. In this study, we evaluated the effects of venestatin, an EF-hand $Ca^{2+}$-binding protein secreted by the parasitic helminth *Strongyloides venezuelensis*, on RAGE activity and immune responses. Our results demonstrated that venestatin bound to RAGE and downregulated the host immune response. Recombinant venestatin predominantly bound to the RAGE C1 domain in a $Ca^{2+}$-dependent manner. Recombinant venestatin effectively alleviated RAGE-mediated inflammation, including footpad edema in mice, and pneumonia induced by an exogenous RAGE ligand. Infection experiments using *S. venezuelensis* larvae and venestatin silencing via RNA interference revealed that endogenous venestatin promoted larval migration from the skin to the lungs in a RAGE-dependent manner. Moreover, endogenous venestatin suppressed macrophage and neutrophil accumulation around larvae. Although the invasion of larvae upregulated the abundance of RAGE ligands in host skin tissues, mRNA expression levels of tumor necrosis factor-α, cyclooxygenase-2, endothelial adhesion molecules vascular cell adhesion protein-1, intracellular adhesion molecule-1, and E-selectin were suppressed by endogenous venestatin. Taken together, our results indicate that venestatin suppressed RAGE-mediated immune responses in host skin induced by helminthic infection, thereby promoting larval migration. The anti-inflammatory mechanism of venestatin may be targeted for the development of anthelminthics and immunosuppressive agents for the treatment of RAGE-mediated inflammatory diseases.

**Funding:** This work was supported by JSPS KAKENHI (grant numbers 18K15140 and 21K06996) to DT and by grants from the Kitasato University Integrative Research Program of the Graduate School of Medical Sciences (https://www.kitasato-u.ac.jp/en/kugsms/index.html) to DT. The funders had no role in the study design, data collection and analysis, decision to publish, or preparation of the manuscript.

**Competing interests:** The authors have declared that no competing interests exist.

## Author summary

Parasitic helminths have evolved smart strategies to thrive in diverse hosts. For example, parasitic helminths secrete various immunomodulators in the host to establish successful tissue migration to their reproductive niche and chronic parasitism. Identification and functional analyses have revealed these immunomodulators may have potential therapeutic effects in the treatment of immune-related diseases. However, few immunomodulators from parasitic helminths have been identified and analyzed to date. In this study, we determined that venestatin, an EF-hand $Ca^{2+}$-binding protein secreted by the parasitic nematode *Strongyloides venezuelensis*, bound to receptor for advanced glycation end products (RAGE), a host pro-inflammatory receptor, which downregulated RAGE-mediated inflammatory responses. *S. venezuelensis* larvae successfully migrated to their niche owing to the anti-inflammatory functions of venestatin. Venestatin could provide a novel therapeutic target for the treatment of RAGE-mediated inflammatory diseases, such as Alzheimer's disease, rheumatoid arthritis, asthma, ulcerative colitis, and diabetes.

## Introduction

Parasitic helminths are highly prevalent worldwide, particularly in developing countries, with over two billion people infected around the globe [1]. Infection with helminths elicits a type 2 immune response, characterized by eosinophilia, mastocytosis, and increased numbers of type 2 helper T cells and serum IgE levels [2–4]. Helminths have evolved effective strategies to regulate the protective immune response in the host [5], thereby establishing successful tissue migration to their reproductive niche and facilitating chronic parasitism. Helminths dampen or overcome the host immune response by secreting a variety of immunomodulators [6,7]. Recently, various immunomodulators were identified from helminth genomes and proteomes, showing potential for immunologic tolerance to both innate and adaptive responses [8]. These immunomodulators may have dual effects on health. On one hand, they may suppress immunological disorders such as allergy and autoimmunity. On the other hand, they may also suppress immune defense mechanisms by antagonizing vaccine efficacy and resistance to microbial infections [9,10].

Percutaneous infection with helminths, such as *Necator*, *Strongyloides*, and *Schistosoma*, induces a robust type 2 immune response in hosts via extensive larval migration [11–13]. However, the innate immune responses to helminths invading the host skin are poorly understood. Helminthic invasion damages host epithelial tissue, and induces the release of host damage-associated molecular patterns (DAMPs) [8], including S100 small calcium-binding proteins, high-mobility group box 1 protein (HMGB1), β-amyloid, and heparin [14]. In healthy cells, HMGB1 acts as a non-histone DNA-binding protein and induces bends in the DNA helix to facilitate interactions between DNA and proteins [15]. DAMPs are recognized by the receptor for advanced glycation end products (RAGE), a multiligand receptor belonging to the immunoglobulin superfamily that is highly expressed in skin cells, including fibroblasts and keratinocytes [16]. RAGE consists of 404 amino acids and contains a V domain followed by C1 and C2 domains, a transmembrane domain, and a short cytosolic tail [17,18]. RAGE/DAMP binding induces the generation of reactive oxygen species (ROS) and activates the signal transduction pathway involving mitogen-activated protein kinases and nuclear factor-kappa B (NF-κB) [14,19]. Subsequently, translocation of NF-κB into the nucleus induces the expression of pro-inflammatory cytokines, such as tumor necrosis factor-α (TNF-α), pro-inflammatory enzymes such as cyclooxygenase 2 (COX2), and endothelial adhesion molecules

such as intercellular adhesion molecule-1 (ICAM-1) and vascular cell adhesion molecule-1 (VCAM-1), thereby triggering inflammatory responses [14,18,20,21]. Although helminthic invasion can be readily sensed by RAGE and cause the accumulation of inflammatory cells in the host skin, larvae achieve migration to their niche before being surrounding by inflammatory cells, particularly during primary infection [22,23]. Moreover, macrophages and dendric cells are also activated by DAMPs from epithelial tissues damaged by helminthic larvae, leading to initiation and amplification of the type 2 immune response. Although various immunomodulators are expressed by parasitic helminths to perturb these immune responses, the relationship between helminthic infection and RAGE is unclear. Helminthic larvae may express immunomodulators that interfere with RAGE-mediated innate and acquired immune responses.

Recently, we identified and characterized an EF-hand $Ca^{2+}$-binding protein, venestatin [24] [DDBJ/GenBank LC189319], from the rodent parasitic helminth *Strongyloides venezuelensis*, a counterpart of the human-infecting species *S. stercoralis*. The infective third stage larvae (iL3s) penetrate the host skin, migrate to the lungs, and grow into lung stage larvae (LL3s); they then reach the small intestine, where they become parthenogenetic adult female worms and produce eggs. Venestatin, a 19.7 kDa soluble protein, is highly conserved in *Strongyloides* spp. and its expression is upregulated after invasion of iL3s. Mature venestatin is secreted from the larvae into host skin tissue; therefore, venestatin may have pivotal roles in the larval migration process. Furthermore, recombinant venestatin (r-venestatin) binds with both mouse and human RAGE [25]. Accordingly, venestatin may act as an immunomodulator in the RAGE signaling pathway, and *Strongyloides* larvae may use venestatin to achieve successful migration.

In the current study, we evaluated the role of venestatin in RAGE-mediated immune responses, larval invasion, and larval migration. Our findings revealed, for the first time, that venestatin inhibited RAGE-mediated immune responses and had beneficial effects on larval migration.

## Results

### Venestatin bound to the C1 domain of RAGE in a $Ca^{2+}$-dependent manner

To confirm the affinity of venestatin for RAGE, recombinant human RAGE [GenPept NP_001340760] or toll-like receptor (TLR) 4 was reacted with immobilized r-venestatin. RAGE, but not TLR4, bound to r-venestatin in a concentration-dependent manner ($K_D$ = 43.6 nM; Fig 1A). Both anti-venestatin and anti-RAGE antibodies inhibited binding between r-venestatin and RAGE (S1 Fig). The RAGE-binding affinity of r-venestatin was comparable to that of other established RAGE ligands, such as glyceraldehyde-bovine serum albumin (Gla-BSA), $N^6$-(carboxymethyl) lysine-BSA (CML-BSA), human HMGB1 [NP_001370341], human S100A6 [NP_055439], and human S100A12 [NP_005612] (Fig 1B). Next, we attempted to determine which RAGE domain interacted with venestatin. The RAGE-binding affinity of r-venestatin was significantly inhibited by C1 and C2 domain-interacting RAGE ligands S100A6 and S100A12 ($p < 0.01$), but not by V domain-interacting ligands Gla-BSA, CML-BSA, and HMGB1 (Fig 1C) [26]. Binding assays with venestatin and recombinant V, C1, or C2 domains revealed that venestatin bound primarily to the C1 domain ($p < 0.0001$) and displayed significant binding with the C2 domain ($p < 0.01$; Fig 1D). A three-dimensional homology model of venestatin was developed using the crystal structure of human multiple coagulation factor deficiency protein 2 (hMCFD2 [PDB 2VRG]) as a template, which has the highest sequence similarity with venestatin (51%). Computational docking using ClusPro 2.0 software suggested that venestatin centrally bound to the C1 domain of RAGE [PDB 3O3U] (Fig 1E). These data

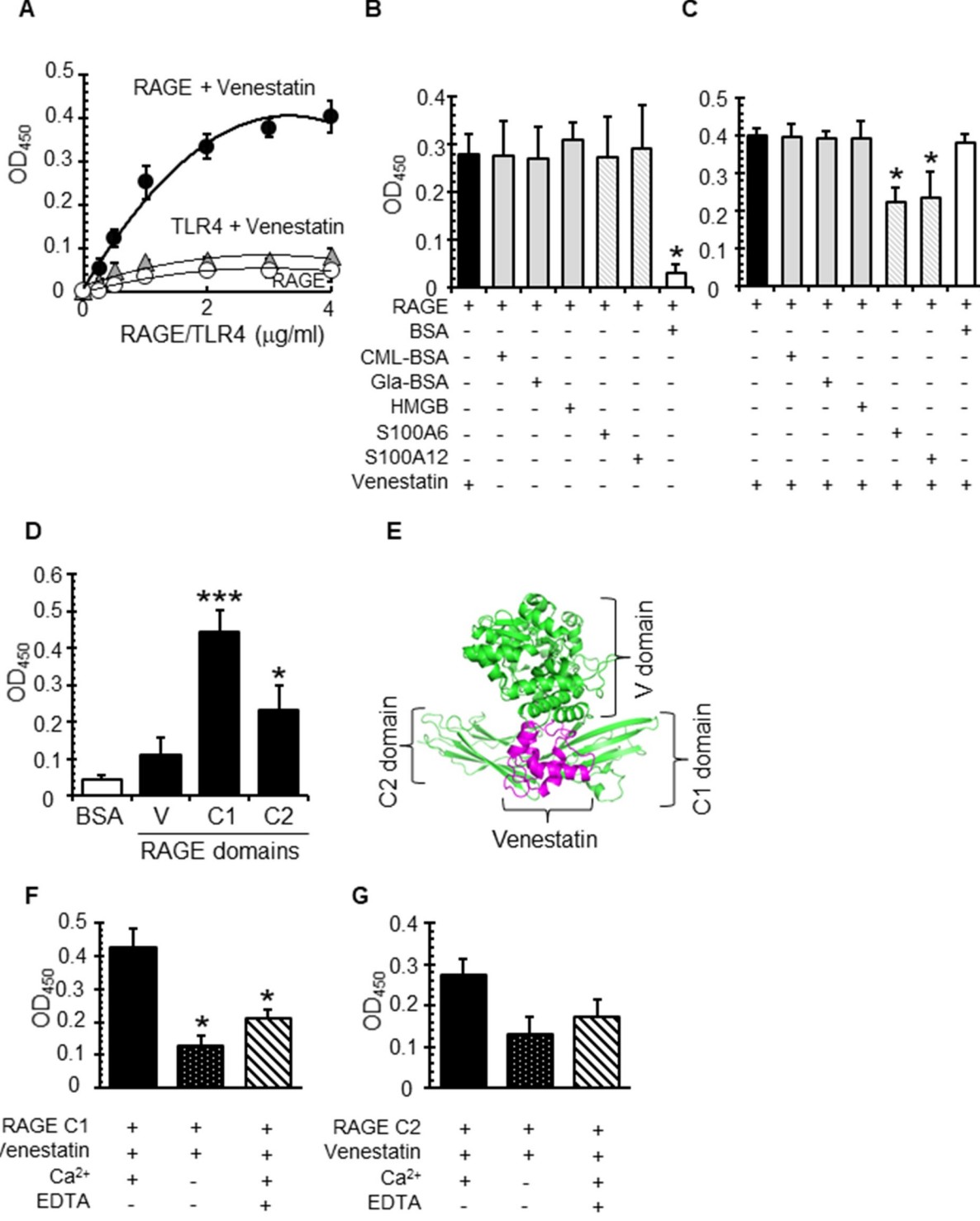

**Fig 1. Venestatin bound to RAGE.** (A) Venestatin bound to RAGE, but not TLR4, in a concentration-dependent manner. Venestatin-coated (4 μg/mL) wells were reacted with RAGE/TLR4 (0–4 μg/mL) and treated with anti-RAGE or anti-TLR4 antibodies. BSA-coated negative control wells were reacted with RAGE (white circles). Antibodies were detected using HRP-conjugated IgG. (B) Venestatin and other established RAGE ligands bound to RAGE. Venestatin, other RAGE ligands, or BSA (4 μg/mL) were used to coat wells and then reacted with RAGE (1 μg/mL). *$p < 0.01$ from the venestatin-RAGE binding group. (C) Competitive binding of venestatin to the C1 and C2 domains of RAGE. RAGE (4 μg/mL) was used to coat wells, and wells were then treated with venestatin alone (8 μg/mL) or a mixture of venestatin plus other RAGE ligands (4 μg/mL). Bound venestatin was detected with anti-venestatin sera. *$p < 0.01$ from a venestatin-RAGE binding group. (D) Venestatin bound to the C1 and C2 domains of RAGE. Wells were coated with V, C1, and C2 domains (4 μg/mL) and then reacted with

venestatin; bound venestatin was then detected. $^*p < 0.01$, $^{***}p < 0.0001$ from the BSA-treated group. (E) Computational docking of venestatin to RAGE. (F) $Ca^{2+}$-dependent C1 domain binding. (G) $Ca^{2+}$-dependent C2 domain binding. Wells were coated with the C1 or C2 domain (4 μg/mL) and reacted with metal-free venestatin (8 μg/mL) with or without $Ca^{2+}$; bound venestatin was then detected. Calcium chelating control was reacted with $Ca^{2+}$ and EDTA. $^*p < 0.01$ from $Ca^{2+}$-containing conditions. Data are expressed as means ± SDs of three independent examinations.

confirmed that venestatin bound mainly to the C1 domain. Because venestatin has two canonical EF-hand $Ca^{2+}$ binding domains and binds RAGE in a $Ca^{2+}$-dependent manner [25], the effects of $Ca^{2+}$ on the binding of venestatin to the C1 and C2 domains were assessed. The binding interaction between venestatin and the C1 domain was significantly stronger in the presence of $Ca^{2+}$ ($p = 0.0029$; Fig 1F). Although the presence of $Ca^{2+}$ did not significantly affect the binding between venestatin and the C2 domain, a $Ca^{2+}$-dependent interaction trend was observed ($p = 0.023$; Fig 1G).

## Venestatin alleviated RAGE-mediated inflammation in mice

To investigate the role of venestatin in the RAGE/ligand axis *in vivo*, we employed mouse inflammation models generated using pro-inflammatory stimulants Gla-BSA and carrageenan, which induce inflammation in RAGE-dependent and -independent manners, respectively [27]. Histopathological examination of the footpad edema model (Fig 2A) revealed that Gla-BSA injection significantly increased the number of infiltrated inflammatory cells (mean ± standard deviation [SD], 370 ± 79 cells/mm$^2$) compared with phosphate-buffered saline (PBS) injection as a negative control (105 ± 43 cells/mm$^2$, $p < 0.0001$). Pretreatment with r-venestatin significantly reduced the number of inflammatory cells (188 ± 53 cells/mm$^2$, $p < 0.0001$) in Gla-BSA injected footpads, but not in carrageenan-injected footpads. Next, the effects of venestatin on mouse pneumonia models were evaluated histopathologically (Fig 2B). Both Gla-BSA and carrageenan induced a massive infiltration of inflammatory cells, thick alveolar walls, and narrow alveolar spaces. Pretreatment with r-venestatin prevented inflammatory lung distortions induced by Gla-BSA, but not by carrageenan. The number of infiltrated cells was significantly lower in the Gla-BSA-induced pneumonia model following r-venestatin pretreatment (279 ± 47 cells/mm$^2$) than in that without r-venestatin pretreatment (504 ± 83 cells/mm$^2$, $p < 0.0001$). Cellular infiltration in both the footpad and lung was not affected by r-venestatin treatment alone (S2 Fig). Production of serum TNF-α, but not IFN-γ, was suppressed by r-venestatin pretreatment in the Gla-BSA-induced inflammation models (S3 Fig). These data suggested that r-venestatin prevented RAGE/ligand axis-mediated inflammation *in vivo*.

## Transcription and translation of endogenous venestatin were suppressed in *S. venezuelensis* larvae by RNA interference (RNAi)

To elucidate the roles of endogenous venestatin secreted from *S. venezuelensis* during its migration process in host animals, we attempted knockdown of venestatin in *S. venezuelensis* larvae via RNAi. Accordingly, LL3s derived from the lungs of *S. venezuelensis*-infected rats were soaked in double-stranded RNA (dsRNA) encoding venestatin (ds*venestatin*). Expression of venestatin mRNA was significantly downregulated in RNAi-treated LL3s in a time-dependent manner, compared with that in control LL3s soaked with firefly luciferase dsRNA (ds*luciferase*); 45% reduction after 24-h soaking, ($p = 0.0007$), 74% reduction after 72-h soaking, ($p < 0.0001$; Fig 3A). Soaking with ds*luciferase* did not affect the expression of venestatin mRNA, indicating that ds*venestatin* specifically disrupted venestatin mRNA. Western blot analysis of the larvae culture medium revealed that the level of secretory venestatin from

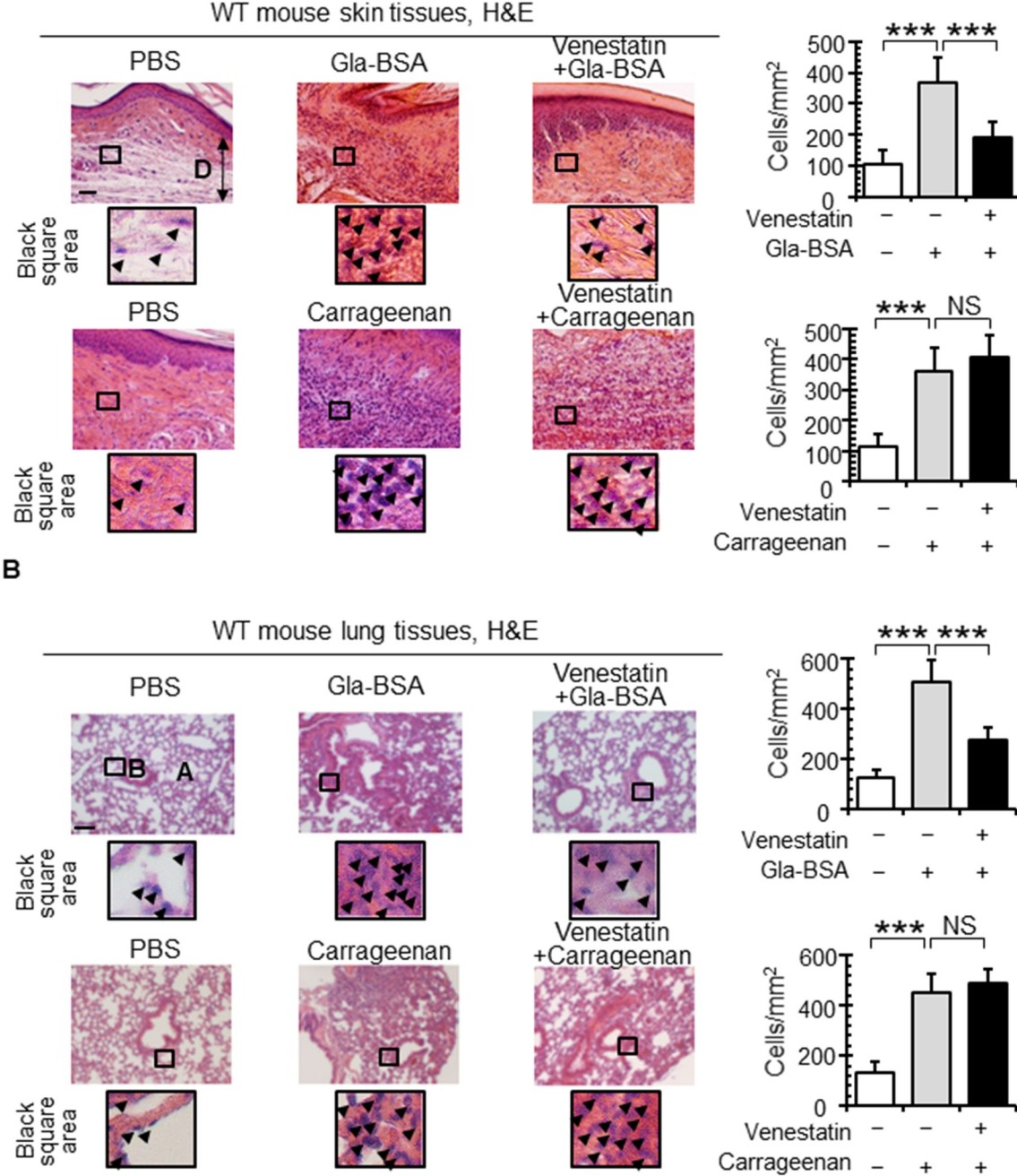

**Fig 2. Venestatin alleviated RAGE/ligand-mediated inflammation.** (A) Venestatin attenuated Gla-BSA-induced mouse footpad inflammation. Venestatin (100 μg) or PBS was injected into the hind footpad of each mouse, followed by injection with Gla-BSA (100 μg), 2% carrageenan, or PBS into the same footpad. After 8 h, the footpads were collected, fixed, and sections were stained with H&E. Black squares indicate magnified insets marking examples of inflammatory cells (arrow heads). White bars indicate the number of cells in PBS controls. D, dermal tissue. (B) Venestatin attenuated Gla-BSA-induced pneumonia in mice. Venestatin (50 μg) or PBS was instilled intranasally, followed by intranasal instillation with Gla-BSA (50 μg), 2% carrageenan, or PBS. After 48 h, the lungs were collected, fixed, and sections were stained with H&E. Black squares indicate magnified insets marking examples of inflammatory cells (arrow heads). White bars indicate the number of cells in PBS controls. A, pulmonary alveolus; B, bronchus lumen. Scale bar: 40 μm. Data are expressed as means ± SDs of 12 fields in two mice. ***$p < 0.0001$. NS, no significant difference.

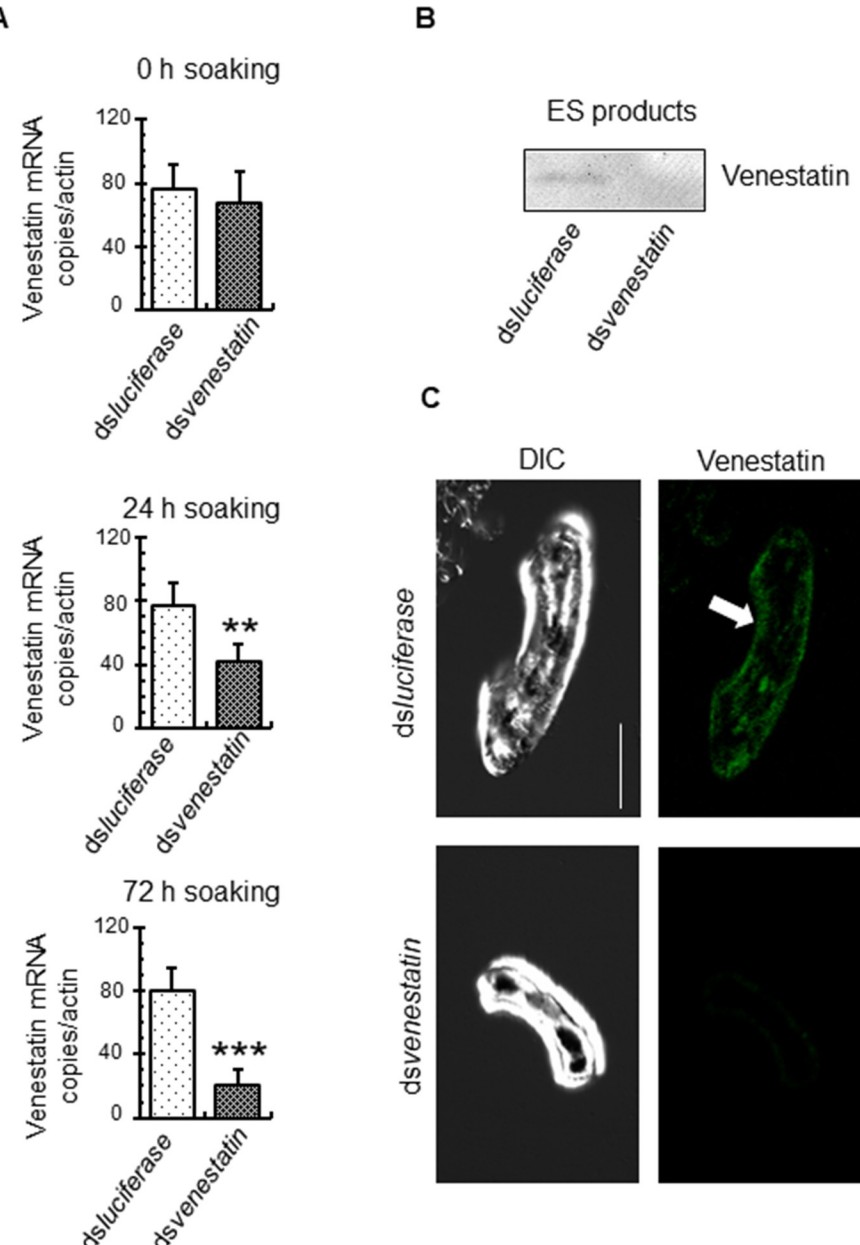

**Fig 3. Post-transcriptional silencing of the venestatin gene in *S. venezuelensis*.** (A) Quantitative RT-PCR analysis of venestatin transcripts. Lung stage larvae (LL3s) of *S. venezuelensis* were incubated with *venestatin* dsRNA as the RNAi-treated group. LL3s in the control group were incubated with *luciferase* dsRNA. The gene encoding *S. venezuelensis* actin-like protein (actin) was used as an internal control, and venestatin mRNA copies/actin was calculated. Data are expressed as means ± SDs for three independent experiments with two technical replicates. $^{**}p < 0.001$; $^{***}p < 0.0001$ from the control group. (B) Effects of gene silencing on venestatin protein expression by western blot analysis. Excretory-secretory (ES) products (1 mL culture medium from 10,000 LL3s) were collected after 72 h of soaking with dsRNA and concentrated to 100 μL by ultrafiltration. Proteins were separated by SDS-PAGE on 12% gels under reducing conditions. Membranes were probed with anti-venestatin sera. (C) *In situ* detection of venestatin expression in LL3s. Immunofluorescent staining of LL3s using mouse anti-venestatin sera was performed after 72 h of soaking with dsRNA. Bound antibodies were probed with green fluorescent-labeled secondary antibodies. The sections were examined by confocal fluorescent microscopy. Differential interference contrast (DIC) images are also shown. Intense expression of native venestatin (arrow) was observed in the hypodermis of LL3s in the control group, but not in the RNAi-treated group. Scale bar: 50 μm.

RNAi-treated LL3s was decreased compared with that from control LL3s (Fig 3B). Although endogenous venestatin was localized in the hypodermis and digestive tracts of control LL3s, venestatin-specific immunofluorescence reactions were nearly absent in RNAi-treated LL3s (Fig 3C). The mobility and morphology of all LL3s were unchanged, despite soaking with dsRNA for 72 h (S4 Fig). These results indicated that RNAi using ds*venestatin* efficiently silenced venestatin-specific mRNA expression and subsequent translation of venestatin. The RNAi-treated LL3s were used as venestatin-knockdown larvae for subsequent animal infection experiments.

## Endogenous venestatin promoted larval migration in a RAGE-dependent manner

A previous study reported that subcutaneously injected LL3s could successfully migrate to the lungs and small intestines in mice and displayed normal maturation and egg production [28]. Therefore, we next aimed to identify the relationship between endogenous venestatin from *S. venezuelensis* larvae and host RAGE during the migration process using animal infection experiments with control and venestatin-knockdown larvae in wild-type (WT) and RAGE-null (RAGE$^{-/-}$) mice. The larvae were administered by subcutaneous (s.c.) inoculation into mice. At 72 h post infection (p.i.), fewer petechial hemorrhages due to larval migration were observed in the lungs of WT mice infected with venestatin-knockdown larvae than in those infected with control larvae (Fig 4A). In infected RAGE$^{-/-}$ mice, the number of petechial hemorrhages was comparable between control and venestatin-knockdown larvae. The worm burden in the lungs of WT mice infected with venestatin-knockdown larvae was significantly reduced by 61% (35.9 ± 16.8 larvae) compared with that in mice infected with control larvae (89.8 ± 32.9 larvae, $p$ = 0.0002; Fig 4B). Moreover, the worm burden was comparable between RAGE$^{-/-}$ mice infected with control and venestatin-knockdown larvae. These data supported the observed differences in the number of petechial hemorrhages among the groups. At 96 h p.i., the worm burden was also reduced in the small intestines of WT mice (44.3 ± 19.6 larvae), but not RAGE$^{-/-}$ mice following infection with venestatin-knockdown larvae, compared with that in mice infected with control larvae (78.7 ± 28.0 larvae, $p$ = 0.0052). The worm burden did not differ significantly in the lungs and small intestines between WT and RAGE$^{-/-}$ mice infected with control larvae. The number of adult worms in the small intestines of infected WT and RAGE$^{-/-}$ mice at 168 h p.i. was comparable to that at 96 h p.i. and the adult worms were able to lay eggs (S5 Fig). Thus, these data suggested that endogenous venestatin served an important role in larval migration from the skin to the lungs via the RAGE/ligand axis. Kinetic analysis of larval migration to the lungs indicated that the worm burden in the lungs was lower at 96 h p.i. than at 72 h p.i. (S6 Fig), and larvae were hardly detected in the lungs of all groups at 144 h p.i. These results suggested that larvae migrate to the small intestines from the lungs regardless of venestatin expression.

Next, larvae remaining in the skin were detected by reverse transcription polymerase chain reaction (RT-PCR) analysis of transcripts of *S. venezuelensis* actin-like protein (*S. venezuelensis* actin [WormBase SVE_0864000]) due to the technical difficulties of collecting and counting larvae in skin tissues. The expression level of *S. venezuelensis* actin-specific mRNA in the skin of WT mice infected with venestatin-knockdown larvae was higher than that in the skin of WT mice infected with control larvae at 24 and 48 h p.i.; in contrast, the expression levels of *S. venezuelensis* actin-specific mRNA did not differ in the skin of RAGE$^{-/-}$ mice infected with control or venestatin-knockdown larvae (Fig 5A). The expression levels of mouse actin-specific mRNA were comparable among skin tissues of all groups. *S. venezuelensis* and mouse actin primers did not cross-react with mouse actin and *S. venezuelensis* actin, respectively. At

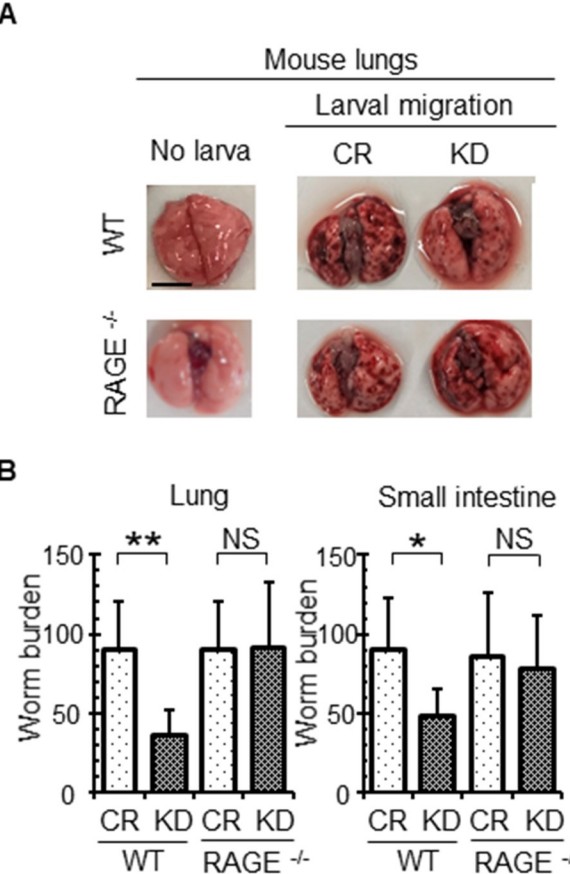

**Fig 4. Effects of post-transcriptional silencing of the venestatin gene on larval migration of *S. venezuelensis*.** (A) Venestatin-specific gene silencing alleviated lung hemorrhage induced by larval migration. Wild-type (WT) or RAGE-null (RAGE$^{-/-}$) mice were infected with 2,000 LL3s treated with ds*luciferase* (CR) or ds*venestatin* (KD). Petechial hemorrhage in the lungs was observed on day 3 (72 h) p.i. Scale bar: 1 cm. (B) Venestatin-specific gene silencing reduced worm burden in the lungs. Lung and small intestinal larval burdens are shown from WT or RAGE$^{-/-}$ mice on days 3 (72 h) and 4 (96 h) p.i., respectively. Data are expressed as means ± SDs of 10 mice from two independent trials. $^*p < 0.01$; $^{**}p < 0.001$. NS, no significant difference.

48 h p.i., only the skin of WT mice infected with venestatin-knockdown larvae contained *S. venezuelensis* actin-specific mRNA. Quantitative transcriptional analysis of *S. venezuelensis* actin demonstrated that the skin of WT mice infected with venestatin-knockdown larvae contained significantly more mRNA than that of WT mice infected with control larvae (24 h p.i., $p < 0.001$; 48 h p.i., $p < 0.01$; Fig 5B). Additionally, mRNA levels did not differ significantly between the skin of RAGE$^{-/-}$ mice infected with control and venestatin-knockdown larvae. These results implied that larvae secreting venestatin favorably migrated out from the host skin in a RAGE/ligand axis-dependent manner.

## Endogenous venestatin suppressed RAGE-mediated immune responses during larval migration

Invasion and migration of control and venestatin-knockdown larvae significantly upregulated the expression of RAGE ligand mRNA, such as HMGB1 [GenBank NM_001313894], S100B [NM_009115], and S100A6 [NM_011313], in mouse skin tissues but did not affect the expression of RAGE [NM_007425] itself (S7 Fig), indicating the larvae may induce RAGE-mediated

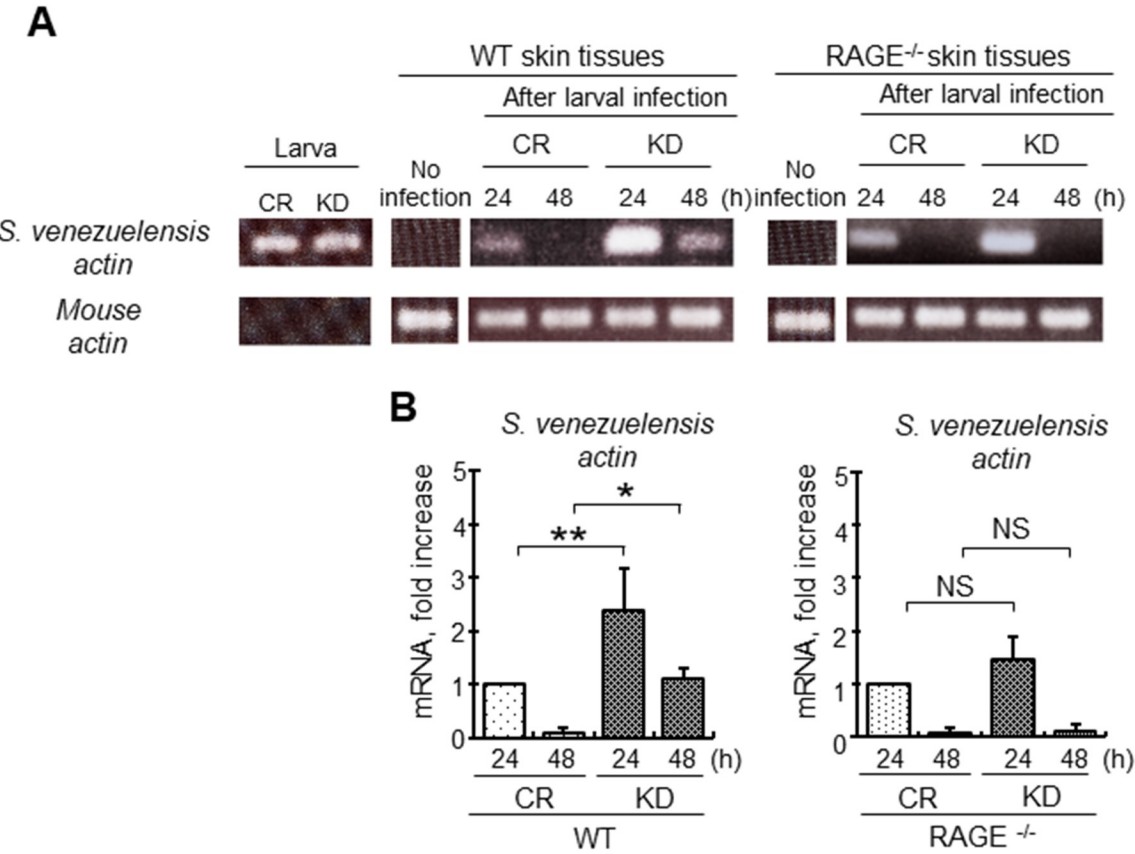

**Fig 5. Post-transcriptional silencing of the venestatin gene inhibited migration of *S. venezuelensis* from skin tissues.** (A) Transcripts of *S. venezuelensis* actin-like protein (*S. venezuelensis* actin) in skin tissue of wild-type (WT) or RAGE-null (RAGE^(-/-)) mice infected with 2,000 LL3s of *S. venezuelensis* treated with ds*luciferase* (CR) or ds*venestatin* (KD) were detected by RT-PCR. Total RNA was extracted from the skin tissues at the larva inoculation site at 24 and 48 h p.i. Naïve skin tissues were used as a negative control (no infection). Expression of mouse β-actin (mouse actin) was shown as an internal control. Expression of *S. venezuelensis* actin in CR or KD LL3s was also assessed. (B) Quantitative RT-PCR analysis of *S. venezuelensis* actin transcripts. Mouse actin was used to normalize the amount of cDNA, and the expression level in skin inoculated with CR LL3s was set as 1. Data are expressed as means ± SDs for three independent experiments with two technical replicates. $^*p < 0.01$; $^{**}p < 0.001$ from the control group. NS, no significant difference.

inflammation responses in host skin. Therefore, we next examined the expression of cytokines, pro-inflammatory enzymes, and endothelial adhesion molecules in skin tissues after invasion by larvae at 6 h p.i. using quantitative RT-PCR analysis. The expression levels of TNF-α [NM_013693] and COX2 [NM_011198] mRNA were significantly upregulated in the skin of WT mice, but not in that of RAGE^(-/-) mice following infection with venestatin-knockdown larvae, compared with those infected with control larvae (TNF-α, $p < 0.01$; COX2, $p < 0.0001$; Fig 6). The mRNA expression levels of type 1 and 2 cytokines, i.e., interferon (IFN)-γ [NM_008337], interleukin (IL)-4 [NM_021283], IL-5 [NM_010558], and IL-13 [NM_008355], were not altered by larval infection. In contrast, expression levels of mRNA encoding endothelial adhesion molecules, i.e., VCAM-1 [NM_011693], ICAM-1 [NM_010493], and E-selectin [NM_011345], were significantly upregulated in the skin of WT mice, but not in that of RAGE^(-/-) mice following infection with venestatin-knockdown larvae, compared with those infected with control larvae ($p < 0.0001$; Fig 7). These data suggested that endogenous venestatin inhibited RAGE-mediated transcription of pro-inflammatory molecules.

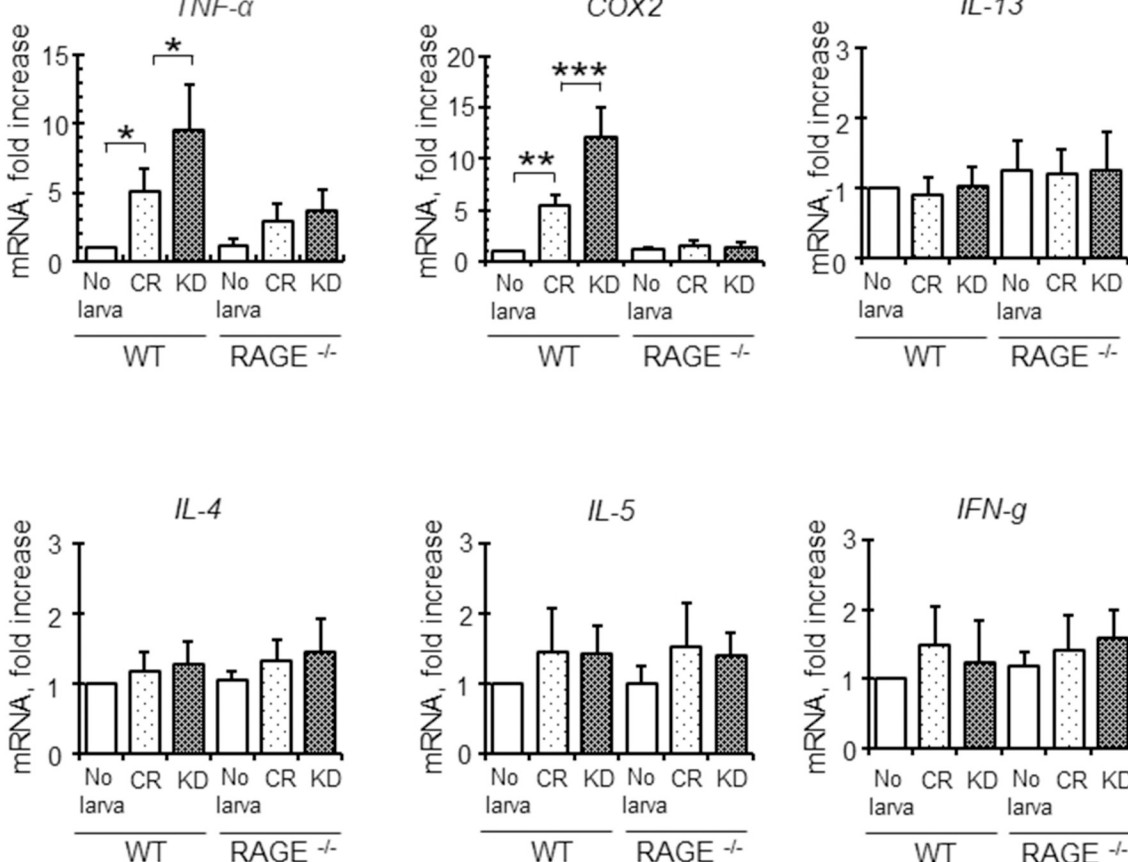

**Fig 6. Effects of endogenous venestatin on cytokine production in mouse skin tissues.** Quantitative RT-PCR analysis of pro-inflammatory molecules from skin tissues of wild-type (WT) or RAGE-null (RAGE$^{-/-}$) mice was performed. Total RNA was extracted from mouse skin tissues at the larva inoculation site at 6 h p.i. with 2,000 LL3s treated with ds*luciferase* (CR) or ds*venestatin* (KD). The mouse GADPH gene was used to normalize the amount of cDNA, and the expression level in naïve skin (no larva) was set as 1. Data are expressed as means ± SDs of three independent experiments with two technical replicates. $^*p < 0.01$; $^{**}p < 0.001$; $^{***}p < 0.0001$.

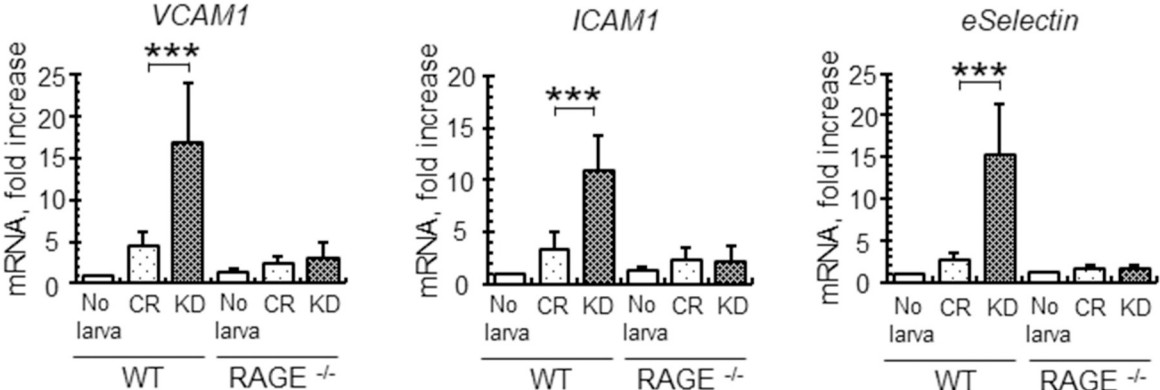

**Fig 7. Effects of endogenous venestatin on adhesion molecule production in mouse skin tissues.** Quantitative RT-PCR analysis of adhesion molecules from skin tissues of wild-type (WT) or RAGE-null (RAGE$^{-/-}$) mice was performed. Total RNA was extracted from mouse skin tissues at the larva inoculation site at 6 h p.i. with 2,000 LL3s treated with ds*luciferase* (CR) or ds*venestatin* (KD). The mouse GADPH gene was used to normalize the amount of cDNA, and the expression level in naïve skin (no larva) was set as 1. Data are expressed as means ± SDs of three independent experiments with two technical replicates. $^{***}p < 0.0001$.

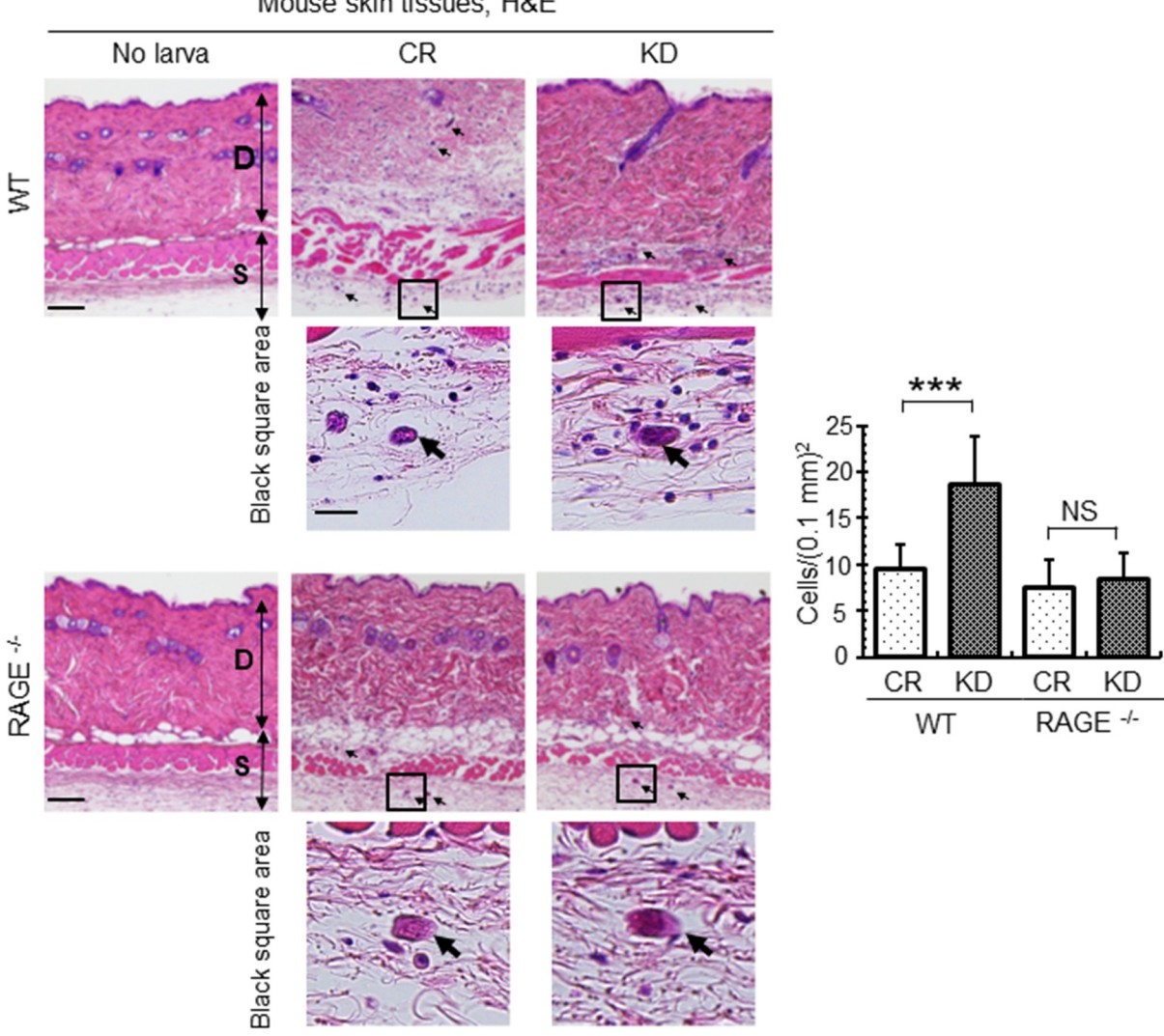

**Fig 8. Endogenous venestatin prevented accumulation of inflammatory cells in mouse skin tissues.** Histochemical analysis of skin tissues from wild-type (WT) or RAGE-null (RAGE$^{-/-}$) mice was performed. The skin tissues were collected from the larva inoculation site at 6 h p.i. with 2,000 LL3s treated with ds*luciferase* (CR) or ds*venestatin* (KD). The sections were subjected to H&E staining, and cells around the larvae (arrows) were then counted. Black square: high-magnification image around larvae. D, dermal tissue; S, subdermal tissue. Scale bar: 40 μm. Data are expressed as means ± SDs of 12 fields from two mice. ***$p < 0.0001$. NS, no significant difference.

Finally, we performed histochemical analyses of mice skin after infection with control or venestatin-knockdown larvae. Cross-sections of larvae were observed in the dermal and subdermal tissues (Fig 8), revealing a substantial accumulation of inflammatory cells around venestatin-knockdown larvae, but not around control larvae in WT mice (Fig 8, black square). The accumulation of inflammatory cells differed significantly between mice infected with control larvae (9 ± 3 cells/0.1 mm$^2$) and venestatin-knockdown larvae (19 ± 5 cells/0.1 mm$^2$, $p < 0.0001$). However, the skin of infected RAGE$^{-/-}$ mice exhibited a slight accumulation of inflammatory cells around both control and venestatin-knockdown larvae. Next, we attempted to identify the types of inflammatory cells around the larvae via immunohistochemistry. Significantly higher numbers of anti-F4/80-specific macrophages and anti-myeloperoxidase (MPO)-specific neutrophils were detected in skin tissues around venestatin-knockdown larvae

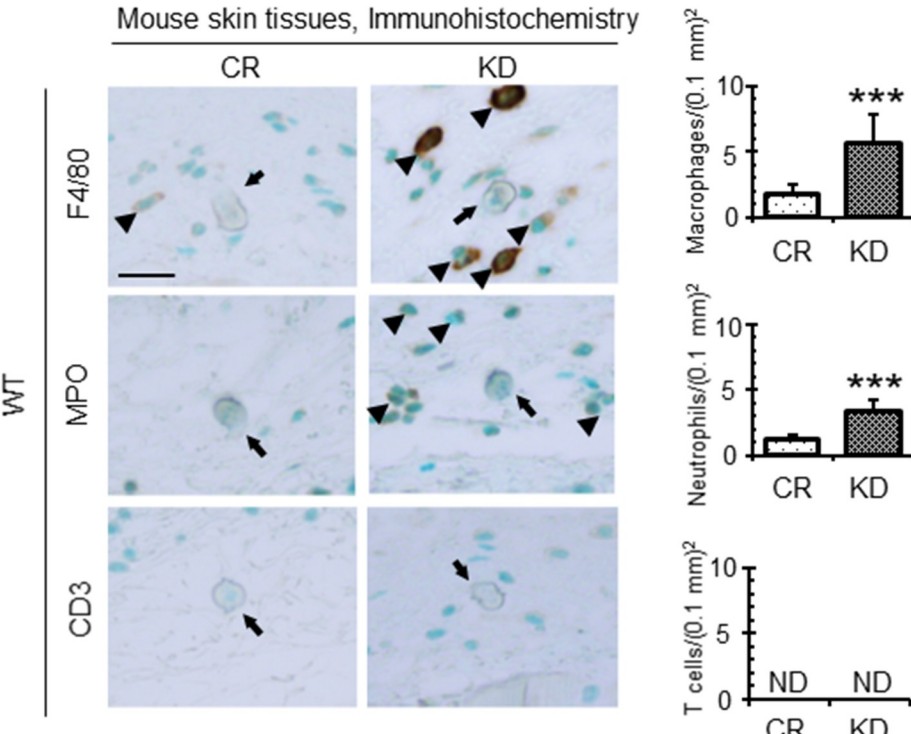

**Fig 9. Endogenous venestatin suppressed the accumulation of macrophages and neutrophils around *S. venezuelensis*.** Immunohistochemical analysis of skin tissues from wild-type (WT) mice was performed. Skin tissues were collected from the larval inoculation site at 6 h p.i. with 2,000 LL3s treated with ds*luciferase* (CR) or ds*venestatin* (KD). The sections were subjected to immunostaining using anti-F4/80 (macrophages), anti-MPO (neutrophils), or anti-CD3 (T cells) antibodies, and positive cells (arrow heads) around the larvae (arrows) were then counted. Scale bar: 25 μm. Data are expressed as means ± SDs of 12 fields from two mice. ***$p < 0.0001$. ND, no detection.

compared with that around control larvae ($p < 0.0001$; Fig 9). Anti-CD3-specific T cells were barely detected around the larvae, nor did anti-B220-specific B cells or anti-IL-5 receptor (IL-5R)-specific eosinophils accumulate around the larvae (S8 Fig). These results indicated that endogenous venestatin suppressed RAGE-mediated accumulation of macrophages and neutrophils around the larvae invading into host skin tissues. Based on these findings, we propose that venestatin induces an immune suppression response by binding to RAGE in the host skin (Fig 10).

## Discussion

EF-hand Ca²⁺-binding proteins have been identified in various organisms, including bacteria, protozoa, helminths, arthropods, and mammals [29–31]. EF-hand proteins regulate calcium signaling in the cytosol to facilitate various cellular functions in both vertebrates and invertebrates [32–34]. Some EF-hand proteins are secreted from cells and exert critical extracellular functions [35–37]. Although there is very limited information regarding secretory EF-hand Ca²⁺-binding proteins in parasitic helminths, we previously identified and cloned a full-length cDNA encoding venestatin, a secretory Ca²⁺-binding protein having two EF-hand motifs from *S. venezuelensis* [24]. Venestatin homologs (83 orthologs) were identified from 69 nematode species (78 genome assemblies), including animal and plant parasitic nematodes, as well as free-living nematodes in WormBase Compara (http://parasite.wormbase.org) [24]. The wide distribution of venestatin orthologs in the phylum Nematoda indicates their crucial function

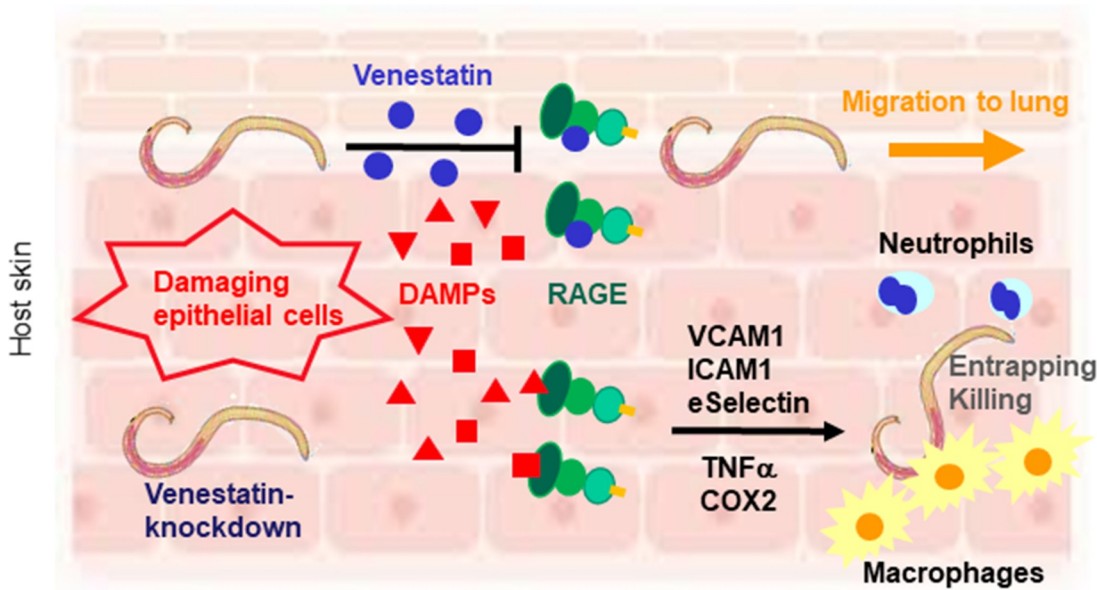

**Fig 10. Schematic diagram showing the roles of venestatin during larval migration of the nematode *S. venezuelensis*.** As RAGE ligands, damage-associated molecular patterns (DAMPs) are released from damaging skin tissues by invasion of *S. venezuelensis* larvae and then bind to RAGE. RAGE signaling induces the expression of cytokines and adhesion molecules, resulting in recruitment of macrophages and neutrophils to entrap or kill venestatin-knockdown larvae. Venestatin downregulates inflammatory responses to control larval invasion. Venestatin may prevent binding between DAMPs and RAGE. Consequentially, larvae secreting venestatin migrate to the lung.

in the nematode life cycle. In the current study, we demonstrated a novel parasitism mechanism through which venestatin suppressed RAGE-mediated immune responses in host skin, leading to successful larval migration. Venestatin orthologs may equip parasitic nematodes with specific functions during their evolution to adapt to the host skin environment.

In the present study, we determined that r-venestatin bound to the C1 and C2 domains of RAGE, with comparable binding affinity to that of other established RAGE ligands. Furthermore, the RAGE C1 domain was found to be the predominant binding site of venestatin, which was dependent on $Ca^{2+}$ ions. The RAGE extracellular domain is composed of a V type immunoglobulin-like domain (V) followed by two C type domains (C1 and C2) [38]. The V domain is the binding domain for most RAGE ligands, which induces downstream signaling [39]. Notably, however, few ligands have been shown to interact with either the C1 or C2 domains [40,41]. Among these, S100A6 and S100A12 have functional roles in cell proliferation and inflammation [42,43]. Characterized by two $Ca^{2+}$-binding EF-hand motifs connected by a central hinge region [44], the presence of $Ca^{2+}$ enhances the binding affinity of S100 proteins with RAGE [45]. Venestatin may specifically affect RAGE-mediated signaling via C1 and C2 domain-interacting RAGE ligands, but not V domain-interacting ligands. However, this possibility is highly unlikely as our data indicated that r-venestatin alleviated inflammation induced by the V domain-interacting ligand Gla-BSA *in vivo*. Binding between S100A6 and the RAGE C1 domain induces a dimeric conformation in RAGE that enables efficient signal transduction [46]. Thus, the C1 domain-binding venestatin may interfere with RAGE homodimerization, resulting in attenuation of V domain-mediated signaling. Although the function of the C2 domain against V domain-mediated signaling is unclear, our data indicates that the C2 domain could also interact with venestatin in a $Ca^{2+}$-dependent manner. Functional analysis of the C2 domain in RAGE structural biology warrants further investigation.

Mouse footpad edema and pneumonia models revealed that purified r-venestatin significantly suppressed inflammation induced by the RAGE ligand Gla-BSA *in vivo*. In contrast, carrageenan-induced inflammation was not alleviated by pretreatment with r-venestatin. We previously reported that carrageenan can induce inflammatory pathology, regardless of whether RAGE is expressed [27]. Our results indicated that venestatin functioned as a RAGE antagonist. Few RAGE antagonists have been identified from natural organisms, and venestatin is the first biological RAGE antagonist isolated from parasitic helminths. Many antagonistic chemical components targeting RAGE have been developed to date, almost all of which interact with the V domain [47]. Importantly, only a few RAGE antagonists have been evaluated in clinical trials, and the results have been inconclusive [48]. Thus, the crystal structure of the complex formed between venestatin and the C1 and C2 domains of RAGE will be valuable for designing novel RAGE antagonists.

Gene silencing of venestatin in *S. venezuelensis* larvae was performed using RNAi to elucidate the role of endogenous venestatin during the migration process in the host. The use of RNAi in parasitic nematodes is known to be challenging, and the specific RNAi effector components associated with RNAi susceptibility are unknown [49]. A recent study reported that RNAi-mediated knockdown in *S. ratti*, a rodent parasitic nematode, could be successfully achieved by soaking with small interfering RNA (siRNA) [50]. Most animal parasitic nematodes, including *Strongyloide* spp., lack the genes required for dsRNA uptake, such as *sid-1* and *sid-2* [49,50], suggesting that siRNA may be better suited for RNAi-mediated knockdown than dsRNA. However, knockdown of specific genes has been achieved for some species of animal [51–53] and plant [54,55] parasitic nematodes by soaking with dsRNA, suggesting alternative dsRNA uptake proteins [56] and endocytotic dsRNA uptake processes [57] in parasitic nematodes. In the current study, we demonstrated that a soaking protocol using ds*venestatin* efficiently silenced venestatin-specific mRNA expression and subsequent translation of venestatin in *S. venezuelensis* LL3 larvae. A pilot experiment determined that knockdown of venestatin was not successful in iL3s (S9 Fig). iL3s reportedly close their mouths and can survive for a long time without feeding [58], until suitable environmental conditions are provided by the host to open their mouths and resume feeding. For *Ancylostoma caninum* iL3s, resumption of feeding and release of secretory proteins is induced by incubation in RPMI 1640 tissue culture medium including canine serum and *S*-methyl-glutathione for 24 h at 37°C [59]. Incubation with serotonin stimulates pharyngeal pumping of *Steinernema carpocapsae* iL3s, leading to uptake of dsRNA to facilitate RNAi [60]. *S. venezuelensis* iL3s may be activated and lead to successful RNAi using these *in vitro* incubation techniques.

Here, we demonstrated that silencing of the venestatin gene in larvae interfered with migration out of the host skin in WT mice, but not in RAGE$^{-/-}$ mice. Furthermore, venestatin-knockdown larvae that reached the lungs could migrate to the small intestine, mature into adult worms, and lay eggs, similar to control larvae. Thus, endogenous venestatin from larvae acted on RAGE to promote migration out of the mouse skin, but may not have affected other processes involved in migration, development, and reproduction of worms. These findings were consistent with those of our previous studies in which anti-venestatin serum interfered with larvae migration from the skin to the lungs, but not from the lungs to the small intestine, and with the observation that r-venestatin interacted with mouse endogenous RAGE [24,25]. Larval invasion into mouse skin tissues induced increased levels of some RAGE-ligands, including HMGB1, S100B, and S100A6, thus the RAGE/ligand axis was assumed to have triggered inflammatory responses in skin tissues after larval invasion. Indeed, the transcripts of pro-inflammatory cytokine TNF-α, pro-inflammatory enzyme COX2, and endothelial adhesion molecules VCAM-1, ICAM-1, and E-selectin were markedly upregulated in RAGE-expressing skin tissues invaded by venestatin-knockdown larvae. The interaction of RAGE-

ligands with RAGE induces the expression of pro-inflammatory cytokines, pro-inflammatory enzymes, and adhesion molecules thorough activation of NF-κB [20]. Our findings suggest that silencing of venestatin may improve the environment into which inflammatory cells infiltrate. Furthermore, our histochemical analysis revealed massive accumulation of inflammatory cells around larvae following silencing of venestatin in the presence of RAGE. Taken together, our results strongly suggest that endogenous venestatin from larvae suppress the RAGE-mediated inflammatory responses of the host, thereby promoting migration out of the host skin. Alarmin cytokines, such as IL-25 and IL-33, from bronchial and intestinal epithelial tissues damaged by helminthic larvae activate type 2 innate lymphocytes (ILC2s), which abundantly secrete IL-5 and IL-13, thereby inducing the type 2 immune response [61]. ILC2s were not activated in skin tissues under our current experimental conditions (6 h p.i.), indicated by IL-5 and IL-13 transcript abundance being unaltered by larval invasion. However, the experimental timepoint may have been too short for induction of ILC2-mediated type 2 immunity in skin tissues [62].

The skin-invading larvae of *S. ratti* induce cellular infiltration by macrophages, neutrophils, and eosinophils in mice and rats [63]. Although eosinophilia is a known feature of helminth infection and eosinophils can kill helminth larvae *in vitro*, protective effects against most helminth species are observed only after secondary infection *in vivo* [4]. Indeed, basophils infiltrate and trap skin-invading larvae of the rodent parasitic nematode *Nippostrongylus brasiliensis* after secondary infection, but not primary infection [23]. However, our immunohistochemical results indicated that macrophages and neutrophils accumulated around venestatin-knockdown larvae after primary invasion into WT mouse skin. *S. stercoralis* larvae were killed by macrophages, neutrophils, and complement in previous *in vitro* assays and *ex vivo* mouse models [64]. Moreover, neutrophil extracellular traps (NETs) are released from mouse neutrophils after *S. stercoralis* infection and function to entrap larvae and prevent their movement without killing larvae *in vitro* and *ex vivo* [65]. Our data indicated that the number of larvae migrating to the lungs from the skin was reduced by silencing of the venestatin gene. The larvae could migrate to the small intestines from the lungs, and then larvae were hardly detected in the lungs, regardless of venestatin expression. Eventually, the worm burden of larvae arriving and mature worms settling in the small intestines was reduced by silencing of venestatin. These data imply that migration of some larvae from the skin was completely inhibited, not merely delayed, by silencing of venestatin. Our current findings and those previously reported suggest that venestatin may facilitate skin-invading larvae to escape from entrapment by NETs and assault by macrophages. Binding of RAGE ligands to RAGE on macrophages induces activation of macrophages [19]. However, alternatively activated macrophages (AAMs) more effectively kill larvae than naïve and classically activated macrophages [64]. Because AAMs have anti-inflammatory functions, the effects of venestatin on macrophage activation is an interesting topic in studies of the therapeutic potential of venestatin in RAGE-mediated inflammatory diseases, such as Alzheimer's disease, rheumatoid arthritis, asthma, ulcerative colitis, and diabetes [66–68].

A venestatin homolog was highly conserved in the *Strongyloides* species, *S. stercoralis* [24]. The nematode species, *Necator americanus*, and the trematode species, *Schistosoma mansoni*, also conserve venestatin homologs in the genomic database. However, the function of venestatin homologs from human-infecting helminths, which migrate from the skin to the lungs, remains to be addressed. Although relationships have been reported between RAGE and some bacterial cutaneous infections, such as *Mycobacterium leprae* and *Staphylococcus aureus* [69,70], the biological significance of RAGE is poorly understood in human infectious diseases, particularly its significance with helminthiasis. A previous study demonstrated that RAGE expression was altered in the early stage of *S. mansoni* infection [71]. Our findings support a

general mechanism of helminthic larvae invasion into skin tissues that express high levels of RAGE, enabling larvae to evade host immunity, although this hypothesis requires further research and validation. Elucidation of the functions of venestatin homologs during the skin invasion process is anticipated to contribute to strategies for controlling cutaneous infection by parasitic helminths in the future.

In conclusion, the EF-hand $Ca^{2+}$-binding protein venestatin, secreted by the helminth *S. venezuelensis*, binds to RAGE and downregulates RAGE-mediated inflammatory responses. Our findings indicate that venestatin plays a key role in immune evasion by *S. venezuelensis* larvae, consequently promoting larval migration from the skin to the lungs. Furthermore, the anti-inflammatory mechanism of venestatin may be targeted for the development of anthelminthics and immunosuppressive agents. The study findings warrant further investigation to assess the therapeutic effects of venestatin in RAGE-mediated pathological models.

## Materials and methods

### Ethics statement

Animal experiments were conducted in accordance with the guidelines of the Animal Laboratory Center of Kitasato University School of Medicine, and all efforts were made to minimize suffering. All animal procedures were approved by the Animal Laboratory Center of Kitasato University School of Medicine (permission-numbers 2018–067, 2019–066, 2020–131 and 2021–141).

### Animals

Mice (Charles River Laboratories Japan, Yokohama, Japan) and rats (Japan SLC, Shizuoka, Japan) were housed in individual cages and in a temperature- and humidity-controlled environment with a 12-h dark-light cycle. The animals were given food and water *ad libitum*. Rats and mice were euthanized by carbon dioxide inhalation and cervical dislocation, respectively.

### Parasites

*S. venezuelensis* HH1 was previously isolated [72] and has been maintained in our laboratories (Kitasato University School of Medicine and Faculty of Medicine, University of Miyazaki, Japan) by serial passaging in male 8-week-old Wistar rats. iL3s were prepared using the filter paper method [73] and administered by s.c. injection (30,000 iL3s/rat). LL3s were recovered from rat lungs at 72–75 h p.i. as previously described [28].

### Production of r-venestatin and anti-venestatin antibodies

Production and purification of r-venestatin were performed as previously described [25] using the silkworm baculovirus expression system. For production of anti-venestatin antibodies, female 5-week-old BALB/c mice were immunized by s.c. injection of 50 μg r-venestatin emulsified with TiterMax Gold adjuvant (TiterMax USA, Norcross, GA, USA). A booster immunization was administered 2 weeks after the first immunization.

### Microtiter plate binding assay of venestatin with RAGE

The binding of venestatin with RAGE was evaluated as previously described [27]. Briefly, r-venestatin (4 μg/mL) was coated onto enzyme-linked immunosorbent assay (ELISA) plates (Thermo Fisher Scientific, Waltham, MA, USA) and stored overnight at 4˚C. After blocking, different concentrations of human recombinant RAGE (R&D Systems, Minneapolis, MN, USA) or TLR4 (R&D Systems) (0–4 μg/mL) were added to the wells and incubated with 50 μL

buffer A (50 mM Tris-HCl, pH 7, 10 mM sodium chloride, and 5 mM calcium chloride) at ambient temperature for 1 h. Negative control wells were coated with BSA prior to incubation with human recombinant RAGE. The wells were washed, and bound proteins were incubated with anti-RAGE (1:500; Merck Millipore, Massachusetts, MA, USA) or anti-TLR4 (1:500; Proteintech Group, Rosemont, IL, USA) antibodies at ambient temperature for 1 h. The proteins were then incubated with horseradish peroxidase (HRP)-conjugated IgG and TMB One Solution (Promega, Madison, WI, USA) and the absorbance was measured using a POWERSCAN instrument (DS Pharma Biomedical, Osaka, Japan) at 450 nm ($OD_{450}$). The $K_D$ value was calculated by a Scatchard plot. To compare the binding ability of venestatin with that of other ligands, including Gla-BSA (produced as previously described [74]), CML-BSA (Medical & Biological Laboratories, Nagoya, Japan), human HMGB1, human S100A6, and human S100A12 (Abnova, Taipei, Taiwan), ELISA plates were coated with r-venestatin, RAGE ligands, or BSA (4 μg/mL). Plates were then blocked, and RAGE (1 μg/mL) was added, followed by binding with anti-RAGE antibodies (1:500). Bound RAGE was detected as described above.

## Inhibition of binding by anti-RAGE and anti-venestatin antibodies

ELISA plates were coated with RAGE (4 μg/mL), incubated with anti-RAGE antibodies (1:100), and then treated with venestatin. Alternatively, venestatin (4 μg/mL) was pre-incubated with anti-venestatin sera (1:100), and the mixture was added to RAGE-coated wells. Bound venestatin was reacted with biotin-labelled anti-venestatin antibodies prepared using a Biotin-Labelled Kit-NH$_2$ (Dojindo Laboratories, Kumamoto, Japan), followed by incubation with HRP-conjugated streptavidin (Sigma-Aldrich, St. Louis, MO, USA) and TMB One Solution. Bound venestatin was detected as described above.

## Competitive binding assay

ELISA plates were coated with RAGE (4 μg/mL) and then incubated with venestatin (8 μg/mL) alone or a mixture of venestatin with other RAGE ligands (4 μg/mL) and 50 μL buffer A at ambient temperature for 1 h. Bound venestatin was detected with anti-venestatin sera (1:1000).

## Domain binding assay

Recombinant V, C1, and C2 RAGE domains were expressed using *Escherichia coli* and then purified as previously described [75]. ELISA plates were coated with the domains (4 μg/mL) and then reacted with venestatin. Bound venestatin was detected with anti-venestatin sera. In the $Ca^{2+}$-dependent RAGE domain binding assay, the C1 or C2 domains of RAGE (4 μg/mL) were coated on ELISA wells and then incubated with metal-free venestatin (8 μg/mL), prepared as described previously [27] in buffer A with or without 5 mM calcium chloride. The calcium chelating control was incubated in buffer A with 5 mM calcium chloride and 5 mM ethylenediaminetetraacetic acid (EDTA). Binding with venestatin was then evaluated.

## Computational docking

Model structures of venestatin were built using hMCFD2 [PDB 2VRG] with the SWISS-MODEL program [76]. Computational docking of venestatin to human RAGE [PDB 3O3U] was performed using ClusPro 2.0 software [77]. ClusPro 2.0 was run on docking mode with PDB 3O3U as Receptor and PDB 2VRG as Ligand.

## RNA intervention

For RNAi of the venestatin gene in *S. venezuelensis*, dsRNA was prepared using the T7 Ribo-MAX Express RNAi System (Promega). The sequence encoding venestatin was cloned into T-Vector pMD-20 (Takara Bio, Otsu, Japan). The inserted sequence was amplified by PCR using the primers T7-VeneF7 (5′-TAATACGACTCACTATAGGTTTGGTTGGGTAGGATC AAT-3′) and T7-VeneR388 (5′-TAATACGACTCACTATAGGCTTCTGATGGTAATGGT GGA-3′), containing the T7 promoter sequences at either end. dsRNA complementary to the firefly luciferase gene was used as a negative control. The inserted sequence of luciferase encoded in the pGEM-lucDNA vector (Promega) was amplified by PCR using the primers T7-LucF (TAATACGACTCACTATAGGGCTTCCATCTTCCAGGGATACG) and T7-LucR; (TAATACGACTCACTATAGGCGTCCACAAACACAACTCCTCC), which also contained T7 promoter sequences. dsRNA complementary to the respective DNA inserts was synthesized by *in vitro* transcription using T7 RNA polymerase.

*S. venezuelensis* iL3s or LL3s (1000–2000 larvae/well) were incubated in 48-well plates (Corning, NY, USA) containing 0.2 mL Dulbecco's modified Eagle's medium (DMEM) supplemented with 50 μg/mL penicillin/streptomycin and 0.25 mg/mL dsRNA (ds*venestatin* or ds*luciferase*). The iL3s and LL3s were incubated at 37°C in a humidified atmosphere containing 5% $CO_2$ or 5% $CO_2$ plus 5% $O_2$, respectively. After 24 or 72 h of incubation with dsRNA, the larvae were incubated with DMEM without dsRNA for 24 h under the same incubation conditions. The larvae were then harvested for experimental infection, and total RNA was extracted for RT-PCR analysis. After incubation for 72 h, the culture supernatant was collected and concentrated more than 10-fold using Centrisart (Sartorius, Göttingen, Germany), with a molecular weight cut-off of 10 kDa. Aliquots were subjected to sodium dodecyl sulfate polyacrylamide gel electrophoresis (SDS-PAGE) and western blot analysis. After incubation for 72 h, the larvae were subjected to immunofluorescence staining with anti-venestatin antibodies.

## Infection with venestatin-suppressed larvae

Male 8-week-old WT C57BL/6J mice and RAGE$^{-/-}$ mice were used for experimental infection with *S. venezuelensis* larvae treated with dsRNA. RAGE$^{-/-}$ mice were generated and maintained as previously described [78]. Two thousand LL3s were administered to the mice by s.c. injection. Mice were euthanized at 6, 24, and 48 h p.i. Histochemical analysis was conducted on skin samples at 6 h p.i., while RT-PCR analysis was performed on skin samples at 6, 24, and 48 h p.i. Lungs and small intestines were excised from mice euthanized at 72, 96, and 144 h p.i., and 96 h p.i., respectively, followed by minced using a surgical knife. Larvae were recovered using the Baermann technique as described previously [79] and larvae were counted under a stereomicroscope. Two independent trials of the experimental infection were conducted. At 168 h p.i., adult worms were recovered and counted from the small intestines and fecal egg output (eggs/g feces) was counted.

## Inflammation models

First, either r-venestatin (100 μg) in 0.15 M PBS (pH 7.2, 50 μL) or PBS alone was injected into the hind footpad of a WT mouse. To evaluate the effects of venestatin on RAGE-independent or -dependent mouse footpad edema models [27], 2% carrageenan or Gla-BSA (100 μg) in saline (100 μL) was injected into the same footpad after 1 h. The footpads of saline-injected mice were used as negative controls. Mice were euthanized after 8 h, and footpads were collected, and fixed in 10% formalin in 75 mM phosphate buffer (pH 7.4).

Next, WT mice were intranasally instilled with either r-venestatin (50 μg) in PBS (25 μL) or PBS alone, followed by 2% carrageenan or Gla-BSA (50 μg) in saline (50 μL) after 1 h. The

lungs of saline-instilled mice were used as negative controls. Mice were euthanized after 48 h, and the lungs were collected and fixed.

Serum cytokine levels (TNF-α and IFN-γ) were determined using commercial ELISA kits (FUJIFILM Wako Shibayagi, Gunma, Japan), following the recommendations of the manufacturer.

## RT-PCR and quantitative RT-PCR

Total RNA was extracted from mouse skin tissues using the RNeasy Mini Kit (Qiagen, Valencia, CA, USA) following the manufacturer's instructions. cDNA synthesis from total RNA was performed using the ReverTra-Plus-RT-PCR kit (TOYOBO, Osaka, Japan). A primer pair for *S. venezuelensis* actin-like protein (5′-CCATACGTTGATGGGAAATTGGTAGC-3′ and 5′-C CGTCGAAAACTGTCCTGGACTAAG-3′ [298 bp]) was used. PCR amplification was carried out using 0.1 μg of each cDNA product and oligonucleotides in a final volume of 20 μL. PCR was performed for 2 min at 94°C and for 30 cycles of 98°C for 10 s, 60°C for 30 s, and 68°C for 30 s, followed by a final elongation at 68°C for 5 min. RT-PCR products were subsequently electrophoresed on agarose gels. The expression level of mouse β-actin [NM_007393] was used as an internal control. For exact comparison of expression levels, quantitative RT-PCR was conducted using LightCycler Capillaries (Roche, Basel, Switzerland) with a 20 μL reaction volume containing 10 μL KOD SYBR qPCR Mix (TOYOBO), 0.2 μM each forward and reverse primer, and 1 μL cDNA. The actin-like protein and internal control primer pairs were the same as for RT-PCR analysis. Transcript abundance was measured using a LightCycler 1.5 instrument (Roche Instrument Center AG, Rotkreuz, Switzerland) according to the manufacturer's instructions. PCR amplification was performed for 2 min at 98°C followed by 40 cycles of 10 s at 98°C, 10 s at 60°C, and 30 s at 68°C. Mouse β-actin was used to normalize the amount of cDNA. The data were analyzed using LightCycler Software Version 3.5 (Roche). Expression levels of mouse TNF-α, COX-2, IL-4, IL-5, IL-13, IFN-γ, RAGE, S100A6, S100B, HMGB1, VCAM1, and E-selectin were also analyzed by quantitative RT-PCR. Mouse glyceraldehyde-3-phosphate dehydrogenase (GADPH [NM_001289726]) was amplified according to the same procedures and used to normalize the amount of cDNA. The mouse primers are listed in S1 Table. Each analysis was carried out at least in triplicate with two technical replicates.

For evaluation of RNAi, total RNA was extracted from larvae of *S. venezuelensis* after dsRNA treatment. cDNA was synthesized as described above. A venestatin-specific primer pair (5′-ATTTCCTGGACAACAACCACCTCTTC-3′ and 5′-GGATTTCTGTGAGCTTGAT CTCGTTG-3′ [438 bp]) was used. mRNA abundance was calculated as the number of mRNA molecules per actin-like protein mRNA.

## Western blot analysis

Larvae culture supernatant was subjected to SDS-PAGE under reducing conditions, and proteins were transferred to polyvinylidene difluoride membranes. The membranes were blocked with Blocking One (Nacalai Tesque, Kyoto, Japan) for 1 h at room temperature. The blots were then incubated overnight at 4°C with anti-venestatin antibodies (1:1000) [24] in 5% Blocking One and 0.05% Tween 20 in 0.15 M NaCl and 50 mM Tris-HCl (pH 8.0; TBS-T). The membranes were washed with TBS-T and incubated with an alkaline phosphatase-conjugated secondary antibody (1:5000; AP-conjugated secondary antibodies; Jackson, Carlsbad, CA, USA) for 1 h at room temperature. The membranes were then washed again, and bound proteins were visualized via staining with nitro blue tetrazolium/5-bromo-4-chloro-3-indolyl phosphate (Promega).

## Histochemical examination

Paraffin-embedded sections (5 μm thick) of fixed mouse tissues were prepared and subjected to hematoxylin and eosin staining or immunostaining, as previously described [80]. The number of inflammatory cells per square millimeter in the peribronchial and dermal tissues or per square 100 micrometers around a larval cross-section was analyzed in two sections from each mouse using Image J software (National Institutes of Health, Bethesda, Maryland, USA). For immunohistochemical staining, the sections were treated with one of the following primary antibodies (Abcam, Cambridge, UK): anti-F4/80 (1:1000), anti-CD3 (1:100), anti-MPO (1:100), anti-IL-5R (1:100), or anti-CD45R (B220) (1:100). Immunostaining was performed using the immune-enzyme polymer method with 3,3-diaminobenzidine as the chromogen. The sections were heated in 50 mM sodium citrate buffer (pH 6.0) using in a microwave 3x for 5 min each. Counterstaining was performed with methyl green. The number of immunostained cells around a larval cross-section per square 100 micrometer was also counted.

Larvae were collected after incubation with dsRNA for 72 h and fixed with 10% formalin in 75 mM phosphate buffer (pH 7.4). Paraffin-embedded sections (5 μm thick) were prepared and subjected to immunofluorescence staining. Briefly, the sections were treated with mouse anti-venestatin sera (1:200) and then with secondary antibodies (Alexa Fluor 488 goat anti-mouse IgG (H+L); Invitrogen, Carlsbad, CA, USA). Slides were mounted with VECTA-SHIELD mounting medium containing DAPI (Vector Laboratories, Burlingame, CA, USA) and observed using an LSM710 confocal fluorescence microscope (Carl Zeiss, Oberkochen, Germany). Before treatment with sera, the sections were incubated with 0.1% trypsin solution in 50 mM Tris-HCl (pH 7.5) and 0.1% $CaCl_2$ for 30 min at room temperature. Pre-immune mouse serum (1:200) was used as a control.

## Statistical analysis

Data are presented as means ± standard deviations (SDs). Statistical significance was determined using the Student's $t$ tests with unequal variance or one-way analysis of variance with Tukey-Kramer's test. $P$ values < 0.01 were considered statically significant using Graph Pad 6.0 software (San Diego, CA, USA).

## Supporting information

**S1 Fig. Inhibition of RAGE binding with anti-venestatin or anti-RAGE antibodies.** RAGE-coated wells were treated with or without anti-RAGE antibodies. Venestatin was pre-incubated with or without anti-venestatin antibodies and added the wells. After washing, bound venestatin was detected with biotin-labelled anti-venestatin. Data are expressed as means ± SDs of three independent experiments. $^*p < 0.01$; $^{**}p < 0.001$.
(TIF)

**S2 Fig. Venestatin did not affect cellular infiltration.** Venestatin or PBS was injected into the hind footpad and instilled intranasally each mouse. After 8 or 48 h, the footpads and lungs were collected, respectively, and sections were stained with H&E. Data are expressed as means ± SDs of 12 fields from two mice.
(TIF)

**S3 Fig. Serum cytokines (TNF-α and IFN-γ) in mouse inflammation models.** (A) Mouse footpad edema model. (B) Mouse pneumonia model. Concentrations of cytokines in the serum were measured by ELISA. The minimum detectable concentrations of TNF-α and IFN-γ were 3.58 and 2.05 pg/mL, respectively. Data are expressed as means ± SDs of 3 mouse sera.

ND, no detection.
(TIF)

**S4 Fig. Incubation with dsRNA did not affect the morphology of *S. venezuelensis* larvae.** Infective lung stage larvae (LL3s) of *S. venezuelensis* were incubated with ds*luciferase* or ds*venestatin*. Differential interference contrast (DIC) images of larvae are shown. Scale bar: 50 μm.
(TIF)

**S5 Fig. Venestatin-knockdown adult *S. venezuelensis* showed productive maturation.** Wild-type (WT) or RAGE-null (RAGE$^{-/-}$) mice were infected with 2,000 LL3s treated with ds*luciferase* (CR) or ds*venestatin* (KD). Small intestinal adult worm burden and fecal egg output from WT or RAGE$^{-/-}$ mice at day 7 (168 h) p.i. are shown. Data are expressed as means ± SDs of 6 mice from two independent experiments. *$p < 0.01$.
(TIF)

**S6 Fig. Kinetics of larval migration of *S. venezuelensis* in mouse lungs.** Wild-type (WT) or RAGE-null (RAGE$^{-/-}$) mice were infected with 2,000 LL3s treated with ds*luciferase* (CR) or ds*venestatin* (KD). Lung worm burdens from WT or RAGE$^{-/-}$ mice at days 4 (96 h) and 6 (144 h) p.i. are shown. Data are expressed as means ± SDs of 6 mice from two independent experiments. *$p < 0.01$.
(TIF)

**S7 Fig. Expression of RAGE and RAGE ligands in mouse skin tissues infected with *S. venezuelensis*.** Quantitative RT-PCR analysis of RAGE and RAGE ligands (HMGB1, S100B, and S100A6) from skin tissue of wild-type (WT) mice was performed. Total RNA was extracted from mouse skin tissues at the larva inoculation site at 6 h p.i. with 2,000 LL3s treated with ds*luciferase* (CR) or ds*venestatin* (KD). The mouse GADPH gene was used to normalize the amount of cDNA, and the expression level in naïve skin (no larva) was set as 1. Data are expressed as means ± SDs from three independent experiments with two technical replicates. *$p < 0.01$; **$p < 0.001$ from the no larva group.
(TIF)

**S8 Fig. B cells and eosinophils did not accumulate around *S. venezuelensis* larvae.** Immunohistochemical analysis of skin tissues from wild-type (WT) mice was performed. Skin tissues were collected from the larval inoculation site at 6 h p.i. with 2,000 LL3s treated with ds*venestatin* (KD). The sections were subjected to immunostaining using anti-B220 (B cells) or anti-IL-5R (eosinophils) antibodies. Arrow heads show larval cross sections. Scale bar: 25 μm.
(TIF)

**S9 Fig. Quantitative RT-PCR analysis of venestatin transcripts of *S. venezuelensis* infective L3s (iL3s) incubated with *venestatin* dsRNA.** iL3s in the control group were incubated with *luciferase* dsRNA. The gene encoding *S. venezuelensis* actin-like protein (actin) was used as an internal control, and venestatin mRNA copies/actin was calculated. Data are expressed as means ± SDs for three independent experiments with two technical replicates.
(TIF)

**S1 Table. List of mouse PCR primers.**
(DOCX)

**S1 Data. Data file for the values used to build graphs.**
(XLSX)

## Acknowledgments

We thank the members of the Department of Parasitology and Tropical Medicine, Kitasato University School of Medicine, and The Animal Laboratory Center of Kitasato University School of Medicine for helpful discussions and excellent technical assistance. We would like to thank Editage (www.editage.com) for English language editing.

## Author Contributions

**Conceptualization:** Daigo Tsubokawa, Taisei Kikuchi.

**Formal analysis:** Daigo Tsubokawa.

**Funding acquisition:** Daigo Tsubokawa.

**Investigation:** Daigo Tsubokawa, Jae Man Lee.

**Methodology:** Daigo Tsubokawa, Taisei Kikuchi, Jae Man Lee.

**Resources:** Takahiro Kusakabe, Yasuhiko Yamamoto, Haruhiko Maruyama.

**Validation:** Taisei Kikuchi.

**Writing – original draft:** Daigo Tsubokawa.

**Writing – review & editing:** Daigo Tsubokawa, Taisei Kikuchi, Haruhiko Maruyama.

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
