## [Decision Letter · Decision Letter 0]

12 Mar 2021

Dear Dr. Tsubokawa,

Thank you very much for submitting your manuscript "Venestatin from parasitic helminths interferes with receptor for advanced glycation end products-mediated immune responses to promote larval migration" for consideration at PLOS Pathogens. As with all papers reviewed by the journal, your manuscript was reviewed by members of the editorial board and by several independent reviewers. In light of the reviews (below this email), we would like to invite the resubmission of a significantly-revised version that takes into account the reviewers' comments.

I believe there is concurrence among the reviewers that this paper constitutes a significant contribution to knowledge about immunomodulatory proteins elaborated by parasitic nematodes. Reviewer 3 was particularly effusive in his/her comments to the effect that this is an outstanding study of its kind. However, Reviewers 1 and 2 both bring up significant shortcomings in data gathered in existing experiments (eg. the need for additional controls) or, in the case of Reviewer 2, the need for some significant additional studies (cytokine profiles, worm migration kinetics and flow cytometric analysis of immune effector cells in inflammatory skin lesions) that should be carried out.

I encourage the authors to carefully consider these comments in preparing a substantial revision of the paper.

We cannot make any decision about publication until we have seen the revised manuscript and your response to the reviewers' comments. Your revised manuscript is also likely to be sent to reviewers for further evaluation.

Sincerely,

James B. Lok

Reviews Editor

PLOS Pathogens

P'ng Loke

Section Editor

PLOS Pathogens

Kasturi Haldar

Editor-in-Chief

PLOS Pathogens

orcid.org/0000-0001-5065-158X

Michael Malim

Editor-in-Chief

PLOS Pathogens

orcid.org/0000-0002-7699-2064

I believe there is concurrence among the reviewers that this paper constitutes a significant contribution to knowledge about immunomodulatory proteins elaborated by parasitic nematodes. Reviewer 3 was particularly effusive in his/her comments to the effect that this is an outstanding study of its kind. However, Reviewers 1 and 2 both bring up significant shortcomings in data gathered in existing experiments (eg. the need for additional controls) or, in the case of Reviewer 2, the need for some significant additional studies (cytokine profiles, worm migration kinetics and flow cytometric analysis of immune effector cells in inflammatory skin lesions) that should be carried out.

I encourage the authors to carefully consider these comments in preparing a substantial revision of the paper.

Reviewer's Responses to Questions

**Part I - Summary**

Reviewer #1: Here Tsubokawa et al. demonstrate that venestatin binds to RAGE to evade host detection by suppressing the immune response. Binding experiments, RNAi, and immune assays support this conclusion. I found the work very interesting and novel as they employ various techniques to tease apart the mechanism of venestatin immune suppression. I also believe the conclusions are well supported by the authors experiments and findings.

Reviewer #2: This is an interesting study examining the role of venestatin in the suppression of inflammation, larval migration in animal models of S. venezuelensis infection. Overall, the findings are interesting but scattered.

Reviewer #3: Mechanistic studies of parasitic nematode effectors are lacking, and thorough mechanistic studies are even rarer. This study represents the best and most detailed study of the mechanism of how a nematode effector dampens host immunity, of which I am aware. Frankly, this was a pleasure to read and is precisely the kind of study that needs to be replicated with more nematode effectors. This work builds on previous studies where venestatin was identified and shown to be upregulated during infection as iL3s migrate through the skin. Overall, the methods are well described and well controlled. The paper is well written, and the logical progression of experiments is clear and intuitive. The overarching message is that venestatin binds RAGE and by doing so it dampens host immunity. These conclusions are well-supported by the data presented. There are a few minor points of concern that, if addressed, would improve the quality of the paper. I don’t think that the few additional points of data I suggest below should be required for publication, but they are reasonable suggestions that would improve what has been presented.

**Part II – Major Issues: Key Experiments Required for Acceptance**

Reviewer #1: 1. Include RAGE only and Venestatin only controls for Fig. 1A

2. Include Ca2+ chelating control experiments in Fig. 1F

3. Include details in text or supplemental data demonstrating how widespread venestatin is within the Phylum Nematoda

4. More detailed discussion/interpretation is required for the RNAi specificity for LL3 versus iL3

5. Do the worm actin primers cross-react with mice. Please elaborate as it appears the host response was also used as an internal control. If they cross-react, how do the authors know which species they are monitoring?

6. Fig 2A: add a magnified inset that marks examples of inflammatory cells. Also, are the white bars the PBS counts? this was unclear

7. Fig. 3B should include loading controls.  

8. Fig4B: RAGE mutant versus wt: are these significantly different? interpret? Also, the worm burden is still significant - how does this fit their model?

Reviewer #2: The major comments that need to be addressed are as follows:

1. In the RAGE mediated inflammation models - both foot edema and pneumonia - only cellular infiltrates are examined. No examination of local or systemic pro-inflammatory cytokines or other mediators is examined. This needs to be performed.

2. In the experiments examining larval migration, kinetics of larval migration are not examined. While it is possible that migration is inhibited, it is also possible that migration is delayed and hence different time points post infection need to examined for worm burden etc.

3. In terms of RAGE mediated inflammation of the skin, flow cytometry to examine different immune and non-immune cell types would be useful.

Reviewer #3: I found no major issues.

**Part III – Minor Issues: Editorial and Data Presentation Modifications**

Reviewer #1: 1. Line 69: include reference(s) for secreted immunomodulators

2. Define HMGB1

3. line 147: estimate "the" role

4. line 179: this sentence doesn't make sense. The authors state that soaking did not affect expression in iL3 and LL3s and conclude that soaking is specific to LL3s

5 line 233: authors switch to italics for proteins in contrast to non-italicized earlier. Be consistent.

6. please change "rvenestatin" to "r-venestatin" or just "recombinant venestatin" to make it clearer for the reader

7. lines 438 and 445 and 449: insert space for new paragraph

8. Include details of setting used with ClusPro 2.0

9. line 138: Sequence similarity - include percentage

Reviewer #2: 1. Typos in the manuscript need to be corrected.

2. No discussion of the homologous proteins in either Strongyloides stercoralis or other lung migrating human helminths is done.

3. What is the translational impact of this axis in human infections?

Reviewer #3: The results presented in Figure 1A seem incomplete. Only 4 different concentrations are tested whereas the authors could perform a full curve that would allow them to calculate the strength of binding in the form of KD. Unless the protein is particularly difficult to produce, I can’t see a good reason to limit the data to what is included in the figure.

Another aspect of the data that seemed incomplete is the lack of additional detail on the C2 domain. Figure 1D illustrates significant binding of venestatin to the C2 domain of RAGE (p<0.01), but this was not further explored. Was this binding also calcium-dependent? Including data for C2 in Figure 1F would be interesting and may add to the discussion point on the pathology induced by V domain interactions.

In reading the section on RNAi phenotypes (170-189), I wanted to see this data performed using iL3s. However, I understand the difficulty of RNAi and iL3s and it was helpful that the authors included a brief discussion of this, justifying their use of LL3s for these experiments. I agree with their use of LL3s and I do not think iL3s need to be used for this paper. Moving forward, the authors should consider activating the iL3s in vitro, which may facilitate RNAi in this life stage. Several labs use different techniques for activation of parasitic IJs. Morris et al. 2017 used the addition of serotonin to facilitate RNAi in Steinernema carpocapsae IJs. Hawdon et al. 1999 describes an activation technique for Ancylostoma caninum, but which has been used in other parasitic nematodes as well. Perhaps these methods would allow for efficient RNAi in S. venezuelensis iL3s.

There were a few minor textual issues as well. I did not catch all of them, but here are some suggested corrections:

-Line 55, change to “downregulated…”

Line 274, change to “extracellular functions.”

References:

Morris R, Wilson L, Sturrock M, Warnock ND, Carrizo D, Cox D, et al. (2017) A neuropeptide modulates sensory perception in the entomopathogenic nematode Steinernema carpocapsae. PLoS Pathog 13(3): e1006185.

doi:10.1371/journal.ppat.1006185

Hawdon JM, Narasimhan S, Hotez PJ: Ancylostoma secreted protein 2: cloning and characterization of a second member of a family of nematode secreted proteins from Ancylostoma caninum. Mol Biochem Parasitol 1999, 99:149-165

PLOS authors have the option to publish the peer review history of their article (what does this mean?). If published, this will include your full peer review and any attached files.

Reviewer #1: No

Reviewer #2: No

Reviewer #3: **Yes: **Adler R. Dillman
---

## [Editor Report · Decision Letter 1]

18 May 2021

Dear Dr. Tsubokawa,

We are pleased to inform you that your manuscript 'Venestatin from parasitic helminths interferes with receptor for advanced glycation end products (RAGE)-mediated immune responses to promote larval migration' has been provisionally accepted for publication in PLOS Pathogens.

Best regards,

James B. Lok

Reviews Editor

PLOS Pathogens

P'ng Loke

Section Editor

PLOS Pathogens

Kasturi Haldar

Editor-in-Chief

PLOS Pathogens

orcid.org/0000-0001-5065-158X

Michael Malim

Editor-in-Chief

PLOS Pathogens

orcid.org/0000-0002-7699-2064

This revised manuscripts has many strengths. It provides a solid body of evidence for venestatin as an immune modulator acting as an antagonist of the primary receptor mediating RAGE-related immune responses. There were a number of substantive points raised by the reviewers of the primary submission and the authors have done a very conscientious job of addressing these, in many cases returning to the lab to gather some essential control data. The only substantive request that was rebutted by the authors was the request by Reviewer 2 that they gather flow cytometric data on local skin sites of penetration by infective nematode larvae. The authors rebuttal, with which I concur, was that the isolation of single-cell suspensions from skin is highly problematic and that generating such suspensions in quantities sufficient for flow analysis would be very time consuming and, while representing a potentially fruitful line of inquiry, would not ultimately, be necessary to support the conclusions of the present paper.
---

## [Editor Report · Acceptance letter]

27 May 2021

Dear Dr. Tsubokawa,

We are delighted to inform you that your manuscript, "Venestatin from parasitic helminths interferes with receptor for advanced glycation end products (RAGE)-mediated immune responses to promote larval migration," has been formally accepted for publication in PLOS Pathogens.

Best regards,

Kasturi Haldar

Editor-in-Chief

PLOS Pathogens

orcid.org/0000-0001-5065-158X

Michael Malim

Editor-in-Chief

PLOS Pathogens

orcid.org/0000-0002-7699-2064